# Robust temporal adiabatic passage with perfect frequency conversion between detuned acoustic cavities

Zhao-Xian Chen [1,4], Yu-Gui Peng [2,4], Ze-Guo Chen [3] ✉, Yuan Liu[1], Peng Chen [1], Xue-Feng Zhu [2] ✉ & Yan-Qing Lu [1] ✉

For classical waves, phase matching is vital for enabling efficient energy transfer in many scenarios, such as waveguide coupling and nonlinear optical frequency conversion. Here, we propose a temporal quasi-phase matching method and realize *robust and complete* acoustical energy transfer between arbitrarily detuned cavities. In a set of three cavities, **A**, **B**, and **C**, the time-varying coupling is established between adjacent elements. Analogy to the concept of stimulated Raman adiabatic passage, amplitudes of the two couplings are modulated as time-delayed Gaussian functions, and the couplings' signs are periodically flipped to eliminate temporal phase mismatching. As a result, robust and complete acoustic energy transfer from **A** to **C** is achieved. The non-reciprocal frequency conversion properties of our design are demonstrated. Our research takes a pivotal step towards expanding wave steering through time-dependent modulations and is promising to extend the frequency conversion based on state evolution in various linear Hermitian systems to nonlinear and non-Hermitian regimes.

The past two decades have witnessed remarkable progress in wave manipulation. In passive systems, by reinspecting the contributions of symmetry, artificial gauge field, and topological charges, kaleidoscopic structures have been designed to allow a wide range of applications, including robust wave localization, transportation, and mode conversion[1–4]. In terms of mode conversion, a well-known manifestation of this concept can be seen in the quantum two-level system, where the eigenmodes can be transformed into each other[5]. Leveraging these insights from the quantum multi-level system, exotic phenomena emerge, such as Landau-Zener tunneling[6–10] and rapid adiabatic passage[11,12]. Adiabatic passage processes have been successfully applied for coherent control of multi-level systems[13–15], even beyond the realm of quantum systems. The coupled waveguide system is a well-known platform for studying the evolution of multi-level systems in which the propagation direction replaces the role of time. In linear situations, since the wave equation is stationary, as a corollary, modulation of the waveguide geometries only leads to the redistribution of wave energy or adjustment of the phase, wherein frequency conversion between the initial and final states accompanied by complete energy transfer is absent[16–22]. Frequency conversion necessitates external-field-assisted temporal modulation to provide the energy, while complete energy transfer requires additional sophisticated parameter design[23,24]. State transfer dynamics in continuously and slowly evolving systems cannot realize unitary transfer between eigenstates with different eigenfrequencies. This can be seen from a general two-level model consisting of two detuned cavities with eigenfrequency difference $\Delta f$ and coupling constant $\kappa$; the maximum energy transfer efficiency is limited to $4\kappa^2/(4\kappa^2 + \Delta f^2)$[5,25–27], due to the phase mismatching between states with distinct eigenfrequencies.

[1]National Laboratory of Solid State Microstructures, Collaborative Innovation Center of Advanced Microstructures, and College of Engineering and Applied Sciences, Nanjing University, Nanjing 210093, China. [2]School of Physics and Innovation Institute, Huazhong University of Science and Technology, Wuhan, Hubei 430074, China. [3]School of Materials Science and Intelligent Engineering, Nanjing University, Suzhou 215163, China. [4]These authors contributed equally: Zhao-Xian Chen, Yu-Gui Peng. ✉e-mail: zeguoc@nju.edu.cn; xfzhu@hust.edu.cn; yqlu@nju.edu.cn

To achieve complete energy transfer, pioneering theoretical work proposes a time-switch method, in which the couplings among sites exhibit well-defined fast sign switch series[24]. To date, the realization of cavity state manipulation and effective coupling sign switching in the time domain remains challenging. Thus, experimental demonstration of complete energy transfer between two detuned cavities has not yet been reported.

In this work, we propose a temporal quasi-phase matching (TQPM) method to compensate for the phase mismatching between detuned acoustic cavities and experimentally demonstrate robust and complete acoustical energy transfer. TQPM is performed by enforcing temporally switched couplings, accomplished by electrically controlled relays and time-varying couplers. The results directly demonstrate complete energy transfer between two cavities with different eigenfrequencies. Furthermore, to achieve robust energy transfer, we introduce an intermediary cavity. Although counterintuitive, it effectively mimics stimulated Raman adiabatic passage (STIRAP), where photon frequencies are adjusted to guarantee energy conservation in the robust population transfer between quantum states in atomic physics[13,14]. As shown in Fig. 1a, two incident electromagnetic waves (denoted as $P(t)$ and $S(t)$) couple the states $|1\rangle$ and $|3\rangle$ via the intermediate state $|2\rangle$. The scheme allows robust excitation transfer through adiabatic evolution along the system's zero-energy eigenstate. Our results extend the scope of STIRAP to the case of detuned acoustic cavities, as schematically shown in Fig. 1b. In addition to the robust transfer channel from cavity **A** to cavity **C** realized by simultaneously modulating the couplings' amplitudes and signs, our system with an optimized delay $\Delta t$ functions as a circulator for transient sound waves and can be used as a perfect unidirectional absorber when a proper loss is

introduced in cavity **B**. We anticipate that our approach will allow implementation in various controllable multi-level systems and thus open new ways for transient acoustic energy manipulation.

## Results

### Dynamic coupling for temporal quasi-phase matching

We begin with a model consisting of two detuned acoustic cavities, for which unitary wave transfer is realized by designing the time-varying couplings with programmable electric elements. The experimental setup for the gain-enhanced cavities and dynamic couplings is schematically shown in Fig. 2a. The acoustic cavities (labeled **A** and **B**) are precisely machined with stainless steel and then sealed by acrylic boards. To implement dynamic coupling with different signs, we focus on the first-order resonant mode, which has an antisymmetric dipole-like profile along the height and is shown in Supplementary Section 1 with the sample photograph. By tuning the cavities' sizes and the feedback circuits (connection shown as red lines), the two cavities resonate at $f_A = 1605$Hz and $f_B = 1655$Hz, respectively, with the excitation spectra presented in Supplementary Section 2. To observe sound energy transfer in the time domain, the cavity mode lifetime must be long enough for adiabatic evolution. Thus, we introduce gain with an in-phase feedback electric system and decrease the cavity loss to $\Gamma = 0.8$Hz. The pressures in the two cavities decay as $e^{-2\pi\Gamma t}$ (see Fig. S3 for the fitting). The dynamic mutual coupling $\kappa_{AB}(t)$ between the two acoustic cavities is realized through the double-channel circuits with active electric elements (connection shown as blue lines), including the voltage-controlled amplifiers (VCAs), phase shifters (PSs) and double-pole, double-throw (DPDT) relays (with their functions explained in the Methods). The feedback circuits also include microphones and speakers, sealed inside the cavities to detect and feed the sound. When both the VCAs and DPDTs are fixed, the effective coupling $\kappa_0$ is constant. Although active electric elements are implemented in the circuit, our system is still dissipative. Thus, the dynamic coupling here is Hermitian and is distinct from previous static implementation[28,29].

Once the coupling is switched on, the initially excited sound waves in the two cavities evolve as

$$\frac{dp_A(t)}{dt} = -2\pi\Gamma p_A(t) - i2\pi\kappa_{AB}(t)p_B(t)e^{i2\pi(f_A - f_B)t}, \quad (1-a)$$

$$\frac{dp_B(t)}{dt} = -2\pi\Gamma p_B(t) - i2\pi\kappa_{AB}(t)p_A(t)e^{-i2\pi(f_A - f_B)t}, \quad (1-b)$$

where $p_A$ and $p_B$ are the complex-valued pressure amplitudes. Because the two cavities are detuned by $\Omega_{AB} = |f_A - f_B| = 50$Hz, the phase difference of the sound pressure in the two cavities varies with time. When the coupling is constant, the second terms in the right part of Eq. (1) change sign with time, indicating temporal phase mismatch, which leads to periodic interruption of the sound energy exchange. For example, we initially excite cavity **A** with $f_A$ for $t < 0$, then switch on the stationary coupling of $\kappa_0 = 9.5$Hz and observe the sound evolution. We ignore the damping of the cavities and define the wave transfer efficiency from cavity **A** to **B** as

$$|S_{BA}(t)|^2 = |p_B(t)|^2 / \sum_j |p_j(t)|^2, \quad (2)$$

which is practically adopted for investigating state evolution in mechanical systems[30]. By analyzing the temporally recorded sound waves in Fig. 2b, the maximum of $|S_{BA}(t)|^2$ is only 0.11, consistent with the prediction of $4\kappa_0^2/(\Omega_{AB}^2 + 4\kappa_0^2)$. In addition, the transfer efficiency varies with a period of $T_0 = 18.6$ms, which is

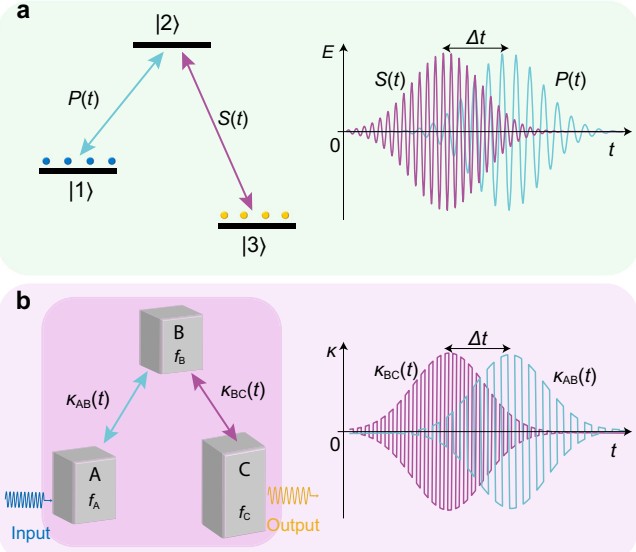

**Fig. 1 | Robust and nonreciprocal adiabatic passage. a** Stimulated Raman adiabatic passage in a three-level system (left panel). With a counterintuitive electromagnetic pulse sequence that $S(t)$ precedes $P(t)$ with $\Delta t$ (shown in the right panel with the vertical coordinate for electric field $E$), the population in $|1\rangle$ can robustly transfer to $|3\rangle$ without transient population in $|2\rangle$. **b** A schematic for the transient acoustic adiabatic passages with three detuned cavities resonant at $f_A$, $f_B$ and $f_C$, respectively (left panel). In addition to the Gaussian-shaped envelops, the signs of the couplings $\kappa_{AB}(t)$ and $\kappa_{BC}(t)$ are periodically flipped to realize temporal quasi-phase mismatching between the waves in adjacent cavities (right panel). The sequential dynamic couplings bring a robust transfer from cavity **A** to cavity **C** and can enable our system to be a circulator or a unidirectional absorber for the transient waves, with the detailed results presented in the following sections.

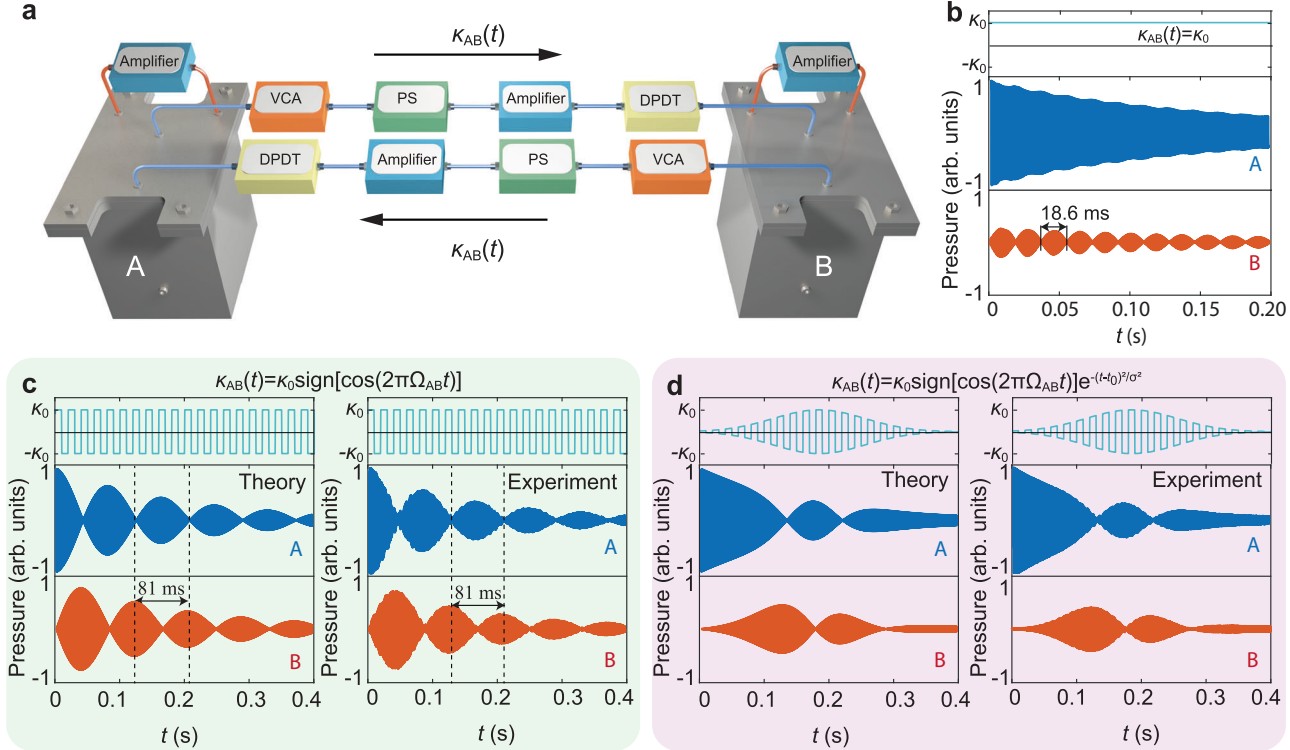

**Fig. 2 | Temporal quasi-phase matching with the periodically changed couplings. a** Simplified experimental setting for two acoustic cavities with dynamic coupling, which is realized by detecting the sound in cavity **A** (**B**) and then feeding it to cavity **B** (**A**) through the power amplifier (connection shown as blue lines). VCAs, PSs, and DPDTs are introduced to modulate the coupling amplitude and sign. Two further amplifiers (connection shown as red lines) are employed to balance the cavity loss and to support a long-lifetime cavity mode. **b** Recorded transient sound waves in two detuned cavities **A** (blue) and **B** (orange) with a constant coupling $\kappa_0$ (cyan line). Due to the phase mismatching, only a small portion of the sound energy in cavity **A** can periodically transfer to cavity **B**. **c–d** Simulated (left panels) and measured (right panels) complete sound wave oscillation between the two cavities with temporal quasi-phase matching condition. The variation of the coupling's amplitude in (**d**) is Gaussian. Source data are provided as a Source Data file.

determined by the Rabi frequency[5], namely $T_0 = 0.5/\sqrt{\Omega_{AB}^2 + 4\kappa_0^2}$. To achieve complete wave oscillation and unitary energy relocation, we propose the TQPM method, which eliminates the consequences of phase mismatch. Specifically, we use square wave voltage variation to control the DPDT relays in the circuit so that the coupling is temporally modulated as

$$\kappa_{AB}(t) = \kappa_0 \text{sign}[\cos(2\pi\Omega_{AB}t)], \tag{3}$$

meaning the sign of the coupling periodically flips with the detuning frequency of $\Omega_{AB} = 50$Hz. Considering the Fourier expansion of $\kappa_{AB}(t)$, namely $\kappa_{AB}(t) = \sum \kappa_{0n} e^{in2\pi\Omega_{AB}t}$, where $n$ is an odd integer, it is apparent that the ±1 order components have the largest amplitude (*i.e.*, $\tilde{\kappa}_0$) and can effectively compensate for the beat frequency in the coupling term in Eq. (1). Significantly, this approach may be considered as the temporal analog to the spatial quasi-phase matching strategy extensively employed in nonlinear optics. This strategy involves meticulously poling nonlinear crystals to counteract phase mismatches among various harmonics[31–33]. By leveraging this technique, one can achieve a broad-spectrum effect, further enhanced through the application of adiabatic passage[34–37]. Under the weak coupling condition, *i.e.*, when $\kappa_0$ is much smaller than $\Omega_{AB}$, retaining the ±1 order Fourier components and neglecting other higher-order series allows Eq. (1) to be written as a Schrödinger-type equation

$$i\frac{d|\psi(t)\rangle}{dt} = \mathbf{H}_{TQPM}|\psi(t)\rangle, \tag{4}$$

where $|\psi(t)\rangle$ represents the pressures of the system and the 2×2 Hamiltonian is

$$\mathbf{H}_{TQPM} = 2\pi \begin{bmatrix} -i\Gamma & \tilde{\kappa}_0 \\ \tilde{\kappa}_0 & -i\Gamma \end{bmatrix}. \tag{5}$$

Apparently, following our design principle, now the two cavities can be effectively coupled like two identical ones with coupling constant $\tilde{\kappa}_0$. When cavity **A** is excited, indicating an initial condition to be described as a superposition of two eigenstates of $\mathbf{H}_{TQPM}$, the amplitude of the sound waves in the two cavities is determined by adiabatic evolution (see the Methods for details). The simulated transient sound pressures are given in the left panel of Fig. 2c, showing that the sound wave energy completely oscillates between the two cavities, *viz.*, $|S_{BA}|^2$ can be unity at some discrete time points. To verify our TQPM theory, we use step variation of the voltage to control the DPDT relays to modulate the coupling according to Eq. (3) after switching off the sound source in cavity **A**. Consistent with the theoretical prediction, we observe complete sound wave oscillation, as shown in the right panel of Fig. 2c. The measured oscillation period is 81ms, from which we can determine the strength of the effective coupling as $\tilde{\kappa}_0 = 6.1$Hz, consistent with the value extracted from the measured spectrum. Compared to the theoretical results obtained by neglecting higher-order terms in the Fourier series, the experimental values vary less smoothly.

The effective coupling $\tilde{\kappa}_0$ does not necessarily need to be constant to realize unitary wave transfer. To mold the two-cavity system into the building block for transient adiabatic passage, the variation of

the coupling's amplitude and sign need to be simultaneously programmed. In addition to flipping the coupling sign with the DPDT relays, we program the VCAs to modulate the coupling strength as a time-dependent Gaussian function (details are presented in Supplementary Section 3). Thus, the dynamic coupling is expressed as

$$\kappa_{AB}(t) = \kappa_0 \, \text{sign}\left[\cos\left(2\pi\Omega_{AB}t\right)\right] e^{-\frac{(t-t_0)^2}{\sigma^2}}. \tag{6}$$

Here, we set $t_0 = 0.185\text{s}$ and $\sigma = 0.1\text{s}$ (see Supplementary Section 4 for the parameter selection). With these initial conditions, both the theoretical and experimental results in Fig. 2d show a complete energy exchange between the two cavities. Obviously, the simultaneous modulation of the effective coupling's amplitude and phase is a powerful tool for achieving temporal adiabatic passages.

## Robust temporal adiabatic passage for sound

Utilizing the TQPM theory, unitary sound energy transfer between detuned cavities becomes achievable. However, the process lacks robustness and is sensitive to the system parameters. An essential step towards achieving robustness is the introduction of an intermediate cavity as a bridge to mediate the wave energy from cavity **A** to the target cavity **C**. In the three-cavity system shown in Fig. 1b, dynamically modulated coupling $\kappa_{BC}(t)$ and $\kappa_{AB}(t)$ are employed to drive the initial state and to realize robust and nonreciprocal adiabatic passage. For generality, we set the first-order resonant frequency of cavity **C** as $f_C = 1585\text{Hz}$ so that all three cavities have different resonant frequencies. We modulate the sign of the dynamic coupling $\kappa_{BC}(t)$ with frequency $\Omega_{BC} = 70\text{Hz}$ according to the TQPM method proposed here. The three-cavity system with dynamic couplings is described by the simplified Hamiltonian

$$\mathbf{H}_{TQPM}(t) = 2\pi \begin{bmatrix} -i\Gamma & \widetilde{\kappa}_{AB}(t) & 0 \\ \widetilde{\kappa}_{AB}(t) & -i\Gamma & \widetilde{\kappa}_{BC}(t) \\ 0 & \widetilde{\kappa}_{BC}(t) & -i\Gamma \end{bmatrix}, \tag{7}$$

where $\widetilde{k}_{AB}(t)$ ($\widetilde{k}_{BC}(t)$) is the effective temporal coupling between cavities **A** and **B** (**B** and **C**). We judiciously program the VCAs and the corresponding amplitudes of the couplings obtain Gaussian modulations as $\widetilde{k}_{AB}(t) = \widetilde{k}_0 e^{-(t-t_0-\Delta t)^2/\sigma^2}$ and $\widetilde{k}_{BC}(t) = \widetilde{k}_0 e^{-(t-t_0)^2/\sigma^2}$, respectively. Apparently, we have $\widetilde{k}_{AB}(t)/\widetilde{k}_{BC}(t) \xrightarrow{t \to 0} 0$ and $\widetilde{k}_{AB}(t)/\widetilde{k}_{BC}(t) \xrightarrow{t \to +\infty} +\infty$ when we set $\Delta t > 0$, namely $\kappa_{AB}(t)$ lags behind $\kappa_{BC}(t)$. According to Eq. (7), the mixing angle, defined as $\theta(t) = \arctan[\widetilde{k}_{AB}(t)/\widetilde{k}_{BC}(t)]$, changes continuously from 0 to $\pi/2$, which determines the mode fields of the eigenstates $|\psi_0(t)\rangle$ and $|\psi_\pm(t)\rangle$ (see Eq. (11) in the Methods).

Specifically, for the case with cavity **A** being initially prepared, the wave dynamic follows the zero-energy state $|\psi_0(t)\rangle$. The sound wave in cavity **A** can robustly and thoroughly transfer to cavity **C** with frequency conversion from $f_A$ to $f_C$. Similar to the definition in Eq. (2), we get the forward wave transfer efficiency as $|S_{CA}|^2 = |p_C(t_{end})|^2 / \sum |p_j(t_{end})|^2$ at $t_{end} = t_0 + 2\sigma + \Delta t$. Obviously, $|S_{CA}|^2$ denotes the fidelity of the zero-energy mode after the evolution. By taking Eq. (7) into the wave coupling equations, we simulate $|S_{CA}|^2$ as a function of $\Delta t$ and $\widetilde{\kappa}_0$. As shown in Fig. 3a, a unitary transfer is achievable in an ample parameter space, demonstrating that the transfer channel is robust because the adiabaticity is well satisfied. For demonstration, we set $\widetilde{\kappa}_0 = 6.1\text{Hz}$ and measure the sound wave transfer from cavity **A**–**C** when $\Delta t$ varies from 0 to 150ms. As shown in Fig. 3b, the energy transfer efficiency (circles) is close to complete, matching the simulated results

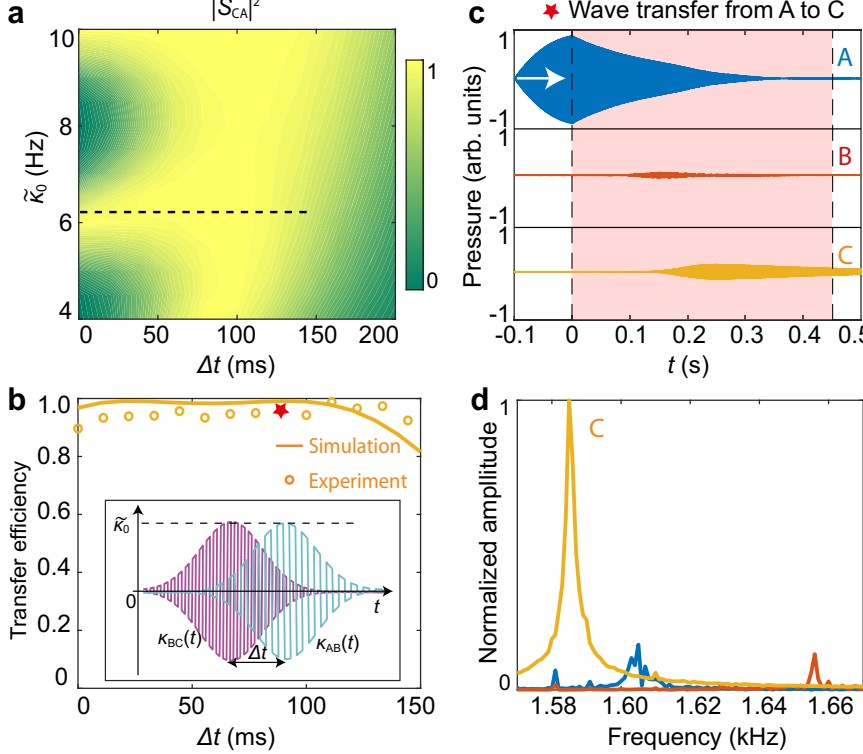

**Fig. 3 | Robust transfer channel based on adiabatic passage. a** Numerically simulated transfer efficiency $|S_{CA}|^2$ as a function of $\Delta t$ and $\widetilde{\kappa}_0$. **b** Simulated (curve) and measured (circles) transfer efficiency $|S_{CA}|^2$ with $\widetilde{\kappa}_0 = 6.1\text{Hz}$, which is denoted in (**a**) with the dashed line. The inset shows the dynamic couplings $\kappa_{AB}(t)$ and $\kappa_{BC}(t)$ with parameters $\widetilde{\kappa}_0$ and $\Delta t$. The red star denotes the case shown in (**c**–**d**) with $\Delta t = 90\text{ms}$. **c** Recorded sound waves in the three cavities. Cavity **a** is initially excited at $f_A$ for $t < 0$, as denoted with the white arrow. **d** Fourier spectra of the transient acoustic pressures in (**c**) with $t > 0.45\text{s}$, demonstrating the complete frequency conversion of the initial sound wave from $f_A$ to $f_C$ after the evolution. Source data are provided as a Source Data file.

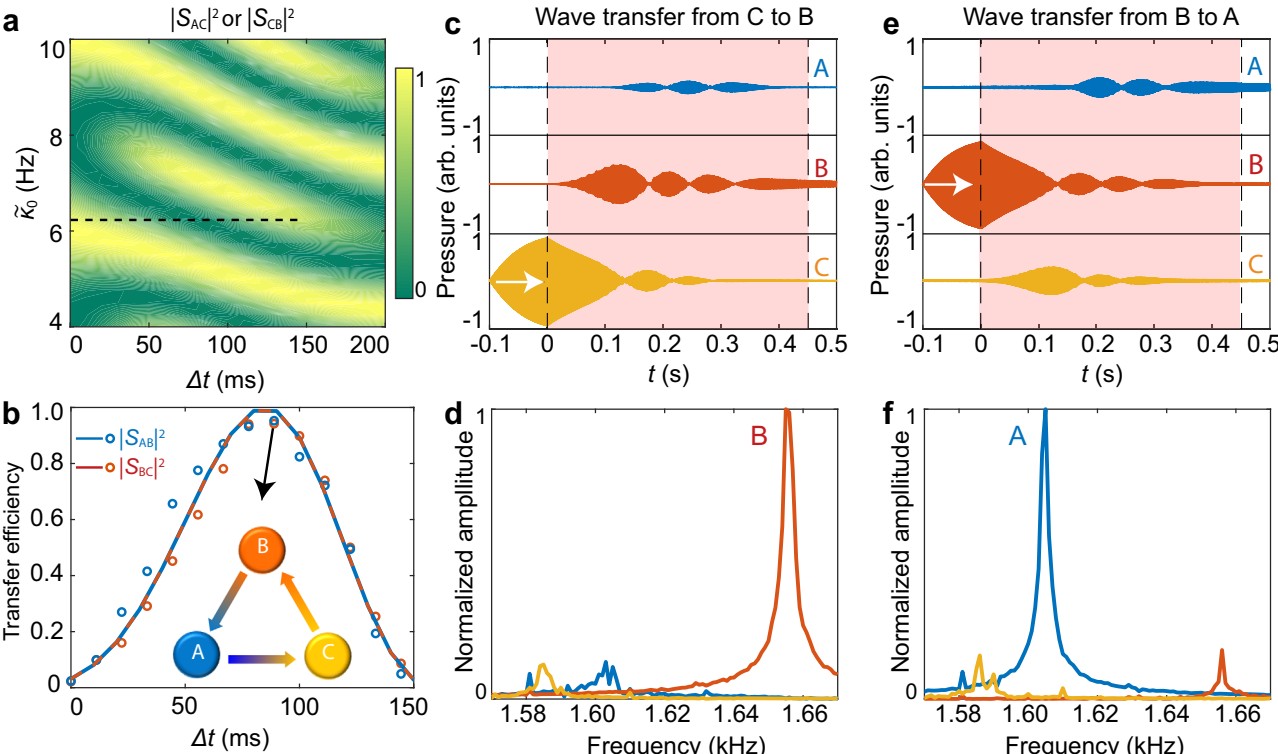

**Fig. 4 | A frequency-converting circulator based on temporal adiabatic passage. a** Numerically simulated wave transfer efficiencies $|S_{AC}|^2$ and $|S_{CB}|^2$ as a function of $\Delta t$ and $\widetilde{\kappa}_0$. The black dashed line denotes the parameter space used in (**b**). **b** Simulated (curves) and measured (circles) transfer efficiencies $|S_{BC}|^2$ and $|S_{AB}|^2$ with $\widetilde{\kappa}_0 = 6.1$ Hz. The inset shows the sound circulations with $\Delta t = 90$ ms, which are given in the following **c**–**f** together with **c**-**d** in Fig. 3. The gradient colors denote the one-way frequency conversions. **c** Recorded sound waves in three

cavities. Cavity **C** is initially excited at $f_C$ for $t < 0$, as denoted by the white arrow. **d** Fourier spectra of the transient pressures in (**c**) with $t > 0.45$ s, showing the frequency conversion effect from $f_C$ to $f_B$. **e** and **f** are similar as (**c**) and (**d**) but with cavity **B** being initially excited at $f_B$, showing the wave transfer from cavity **B** to **A** with frequency conversion from $f_B$ to $f_A$. Source data are provided as a Source Data file.

(solid curves) well. As an example, we choose $\Delta t = 90$ ms and record the transient evolutions of sound waves in Fig. 3c, the corresponding Fourier spectra of the waveform after the transfer ($t > 0.45$ s) are plotted in Fig. 3d. The results show that the sound wave transfers from cavity **A**–**C** along with the frequency conversion from $f_A$ to $f_C$. As predicted by the time evolution of the zero-energy state $|\psi_0(t)\rangle$, only a negligible sound amplitude is observed in cavity **B** (see the Methods and Supplementary Section 5 for the adiabaticity analysis). Notably, the robust sound energy transfer results from the adiabatic evolution of the initially prepared eigenstate $|\psi_0\rangle$. Thus, the sound source must be turned off before dynamic coupling takes place. In addition, though the parameters realized here are suitable for the demonstration purposes, the damping of cavities can be further precisely reduced to increase the transferred sound energy (see Supplementary Section 2 and 4 for the experiment and simulation results).

**Frequency-converting circulator and unidirectional absorber**
In addition to the robust transfer channel from cavity **A**–**C**, our three-cavity system can exhibit strong nonreciprocity since the sequential dynamic couplings break the time-reversal symmetry. As shown in Fig. 4a, the theoretically simulated backward transfer efficiency $|S_{AC}|^2$ varies strongly with variation of both $\Delta t$ and $\widetilde{\kappa}_0$. Meanwhile, we note $|S_{AC}|^2 = |S_{CB}|^2$, which means these two transfer channels share identical dynamic features. For the case with cavity **C** or **B** being initially excited, theoretically, the final energy distributions also relate to $\alpha(t) = \int_0^t \text{real}[\varepsilon_+(t')]dt'$, where $\varepsilon_+$ is the positive eigenvalue of Eq. (7) (see the Methods for details). As a result, the evolution of the wave is determined by both $\Delta t$ and $\widetilde{\kappa}_0$, making them distinct from the forward ones (see Fig. 3a).

We measure the transfer efficiency $|S_{BC}|^2$ ($|S_{AB}|^2$) by initially exciting cavity **C** (**B**) with $f_C$ ($f_B$). As presented in Fig. 4b, the measurements (circles) are consistent with the simulations (curves). In particular, when we set $\Delta t = 90$ ms, the two transfer efficiencies also approach unity, rendering our three-cavity system a counterclockwise sound circulator, which is schematically shown in the inset. In contrast to traditional circulators based on the Faraday effect, the energy circulations in our system are accompanied by unidirectional frequency conversions. In Fig. 4c–f, we present the measured sound wave transfer features and the corresponding spectra with our specified dynamic couplings. Distinct from the forward transfer in Fig. 3c, d, here, all three cavities show oscillations of the energy content before reaching the final localizations. The corresponding spectra in the lower panels show a high frequency-conversion efficiency. Comparatively, without any biased fluid flow or synthetic angular momentum, the adiabatic acoustic passages with dynamic couplings provide a distinct mechanism to realize circulators, particularly for transient sound waves[38–42]. In addition, we can freely tailor the sound frequencies by utilizing the coupling modulation, making our system work as a compact and nonreciprocal sound device.

The transfer details imply one notable feature: adding cavity **B** leads to forming a dark state with no field distribution in **B**. A consequence is that cavity **B** does not even transiently accept energy in the forward transfer process from cavity **A** to **C**, but it does get (and possibly dissipate) power in the backward transfer from cavity **C** to **A**. Taking advantage of this merit, we can build a unidirectional acoustic absorber by introducing loss to the intermediate cavity **B**. With other parameters unchanged, we theoretically simulate the forward transfer efficiency $|S_{CA}|^2$ as a function of the time delay $\Delta t$ and cavity **B**'s loss $\Gamma_B$.

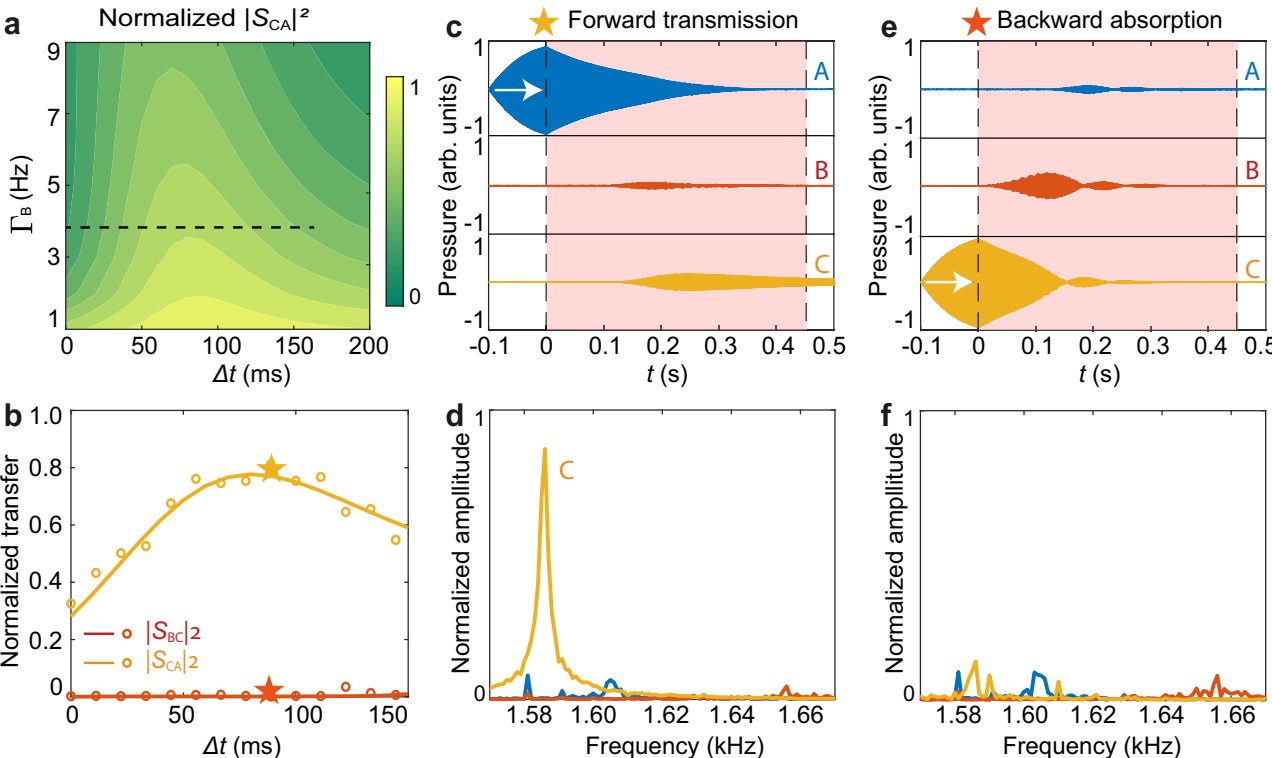

**Fig. 5 | Compact one-way sound absorber. a** Normalized simulated forward transfer efficiency $|S_{CA}|^2$ versus $\Delta t$ and $\Gamma_B$. The black dashed line denotes the parameter range in (**b**). **b** Simulated (curves) and measured (circles) transfer efficiencies $|S_{CA}|^2$ (yellow) and $|S_{BC}|^2$ (red) with the damping rate $\Gamma_B = 4$Hz. The stars represent the cases with $\Delta t = 90$ms in **c–f. c–d** Recorded forward sound wave transfer with cavity **A** being initially excited (**c**) and the corresponding Fourier spectra for the sound waves with $t > 0.45$s (**d**), showing the efficient forward energy transfer and frequency conversion. **e** and **f** are similar to (**c**) and (**d**) but with cavity **C** being initially excited at $f_C$, showing the backward absorption. Here, the transfer efficiencies and spectra are normalized with the forward transfer with $\Gamma_B = 1$Hz and $\Delta t = 90$ms. Source data are provided as a Source Data file.

Here the total energy (the denominator for $|S_{CA}|^2$) is normalized with the case with $\Gamma_B = 1$Hz, and the results are shown in Fig. 5a. Though extra loss is introduced to cavity **B**, we note the forward transfer efficiency $|S_{CA}|^2$ preserves. However, if the sound energy is initially launched in cavity **C**, it will dissipate in **B**. The experimental results shown in Fig. 5b, obtained by increasing the damping of cavity **B** to be $\Gamma_B = 4$Hz, nicely match the theoretical predictions. Distinct from the sound circulator with limited $\Delta t$ selections (see Fig. 4), here, the unidirectional absorber is robust and is almost insensitive to $\Delta t$. In Fig. 5c–f, we provide the measured sound wave evolutions with $\Delta t = 90$ms and the corresponding spectra analysis. For the case with initial excitation in cavity **A**, the relatively large damping of cavity **B** has a negligible influence on the sound transfer, and the normalized $|S_{CA}|^2$ is up to 0.8. By contrast, when cavity **C** is initially excited, the coupling system absorbs and consumes all the energy during the transient coupling.

## Discussion
To conclude, in this work, we have developed a powerful strategy for realizing robust and complete sound wave transfer and efficient frequency conversion between detuned cavities. Notably, such complete energy transfer is achieved by combining the STIRAP concept and TQPM technology, *i.e.*, simultaneously modulating the couplings' amplitudes and signs to realize the transient adiabatic passages of wave energy. Through the external electronically controlled dynamic couplings, the sound state is adiabatically driven in the eigenstate space. Thus, our study provides a versatile and easy-to-implement platform for investigating state evolutions in the time domain. Taking the realization of the frequency-converting sound circulator and unidirectional absorber as examples, we anticipate that our work will bridge the gap between complex state evolution based on the

Schrödinger equation and various wave phenomena based on the Helmholtz equation. When considering the scenario with identical cavities, complete wave oscillation between coupled cavities is naturally satisfied, and the adiabatic passages can be achieved by merely modulating the couplings' amplitudes, presented in Supplementary Section 6. By mimicking the lambda- or ladder-type three-level systems, robust transfer passages between two cavities with arbitrary detuning are achievable (see Supplementary Section 7 for simulations). The generalized adiabatic passages can pave the way for transient sound wave steering[43–46] and can enrich the toolbox of nonreciprocal sound devices[47,48]. In addition, the developed TQPM theory and the generalized STIRAP methodology also shed lights on state manipulation in nano-electromechanical as well as optomechanical systems, which can support multiple mechanical resonances and find more applications in both the classical and quantum realms[30,49,50].

## Methods
### Experimental setup
As schematically shown in Fig. 2a, the effective and mutual coupling is realized by detecting the sound in **A**(**B**) with the microphone and then feeding it to the speaker in **B**(**A**) after amplification (connection shown as blue lines). Here, the VCAs are utilized in the coupling circuit as gates, with which the coupling strength can be freely and temporally programmed through the gate voltage. PSs are adopted to compensate for the phase changes introduced by the VCAs and to achieve real-valued couplings. After the amplifiers, there are DPDT relays, which flip the coupling sign by switching the circuit connection between in-phase and out-of-phase. Two photographs in Supplementary Section 1 show the cavity details and the electric connections for the gain and coupling between two cavities.

For the measurement, 1/4-inch-diameter microphones (Brüel and Kjaer Type 4961) connected to a multichannel analyzer (Brüel and Kjaer Pulse Type 3160) are used to detect the sound pressure in the cavities. The control signals and sound source signals are generated with FeelElec FY8300.

## Adiabatic passages

When the adiabatic condition is satisfied, the sound waves in the coupled cavity system evolve as superpositions of the system's eigenstates, namely

$$|\psi(t)\rangle = \sum_n a_n(0)e^{-i\alpha_n(t)}|\psi_n(t)\rangle, \tag{8}$$

where $n$ indexes the eigenstate, $a_n(0)$ is the initial amplitude of the eigenmode $|\psi_n\rangle$ at $t = 0$, and $\alpha_n(t)$ is the phase angle which is calculated by integrating the corresponding eigenvalues $\varepsilon_n$, viz., $\alpha_n(t) = \int_0^t \varepsilon_n(t')dt'$.

For the two-cavity system described by the simplified Hamiltonian in Eq. (5), we get the eigenvalues $\varepsilon_\pm = -i\Gamma \pm \widetilde{\kappa}_0$ and the corresponding eigenvectors $|\psi_\pm\rangle = (|\varphi_A\rangle \pm |\varphi_B\rangle)\sqrt{2}/2$, where $|\varphi_i\rangle$ denotes the state with complete acoustic energy in cavity $i$. When cavity A is initially excited, the two coupled modes are equally excited, namely $a_\pm(0) = \sqrt{2}/2$. According to Eq. (8), we can directly deduce the transient sound amplitudes of the two cavities as

$$|\psi(t)\rangle = \cos(2\pi\widetilde{\kappa}_0 t)e^{-2\pi\Gamma t}|\varphi_A\rangle - i\sin(2\pi\widetilde{\kappa}_0 t)e^{-2\pi\Gamma t}|\varphi_B\rangle. \tag{9}$$

It is clear that there is complete energy oscillation between the two cavities.

For the three-cavity system with the simplified Hamiltonian in Eq. (7), we get the three eigenvalues

$$\varepsilon_+(t) = -i\Gamma + \sqrt{\widetilde{\kappa}_{AB}^2(t) + \widetilde{\kappa}_{BC}^2(t)}, \tag{10-a}$$

$$\varepsilon_0(t) = -i\Gamma, \tag{10-b}$$

$$\varepsilon_-(t) = -i\Gamma - \sqrt{\widetilde{\kappa}_{AB}^2(t) + \widetilde{\kappa}_{BC}^2(t)}, \tag{10-c}$$

and the corresponding eigenvectors

$$|\psi_+(t)\rangle = \frac{\sin\theta(t)}{\sqrt{2}}|\varphi_A\rangle + \frac{1}{\sqrt{2}}|\varphi_B\rangle + \frac{\cos\theta(t)}{\sqrt{2}}|\varphi_C\rangle, \tag{11-a}$$

$$|\psi_0(t)\rangle = \cos\theta(t)|\varphi_A\rangle - \sin\theta(t)|\varphi_C\rangle, \tag{11-b}$$

$$|\psi_-(t)\rangle = \frac{\sin\theta(t)}{\sqrt{2}}|\varphi_A\rangle - \frac{1}{\sqrt{2}}|\varphi_B\rangle + \frac{\cos\theta(t)}{\sqrt{2}}|\varphi_C\rangle, \tag{11-c}$$

where the mixing angle $\theta(t)$, defined as $\theta(t) = \arctan[\widetilde{\kappa}_{AB}(t)/\widetilde{\kappa}_{BC}(t)]$, determines the field distributions among the cavities. When the time variation of the amplitudes of the two couplings follows Gaussian shape $\widetilde{\kappa}_{AB}(t) = \widetilde{\kappa}_0 e^{-(t-t_0-\Delta t)^2/\sigma^2}$ and $\widetilde{\kappa}_{BC}(t) = \widetilde{\kappa}_0 e^{-(t-t_0)^2/\sigma^2}$ with $\Delta t > 0$, the sequential coupling satisfy $\widetilde{\kappa}_{AB}(t)/\widetilde{\kappa}_{BC}(t) \overset{t\to 0}{\to} 0$ and $\widetilde{\kappa}_{AB}(t)/\widetilde{\kappa}_{BC}(t) \overset{t\to +\infty}{\to} +\infty$, meaning that $\theta(t)$ smoothly changes from 0 to $\pi/2$ through the modulation.

For the initial condition of $|\psi(0)\rangle = |\varphi_A\rangle$, according to the eigenvectors in Eq. (11), we obtain $a_0(0) = 1$ and $a_\pm(0) = 0$, meaning that only the zero-energy state is excited. In this case, the transient state

function $|\psi(t)\rangle$ evolves as the adiabatic evolution of $|\psi_0(t)\rangle$. Ignoring the system's dissipation, the sound intensities in the cavities with $t$ can be given as

$$I_A(t) = |\cos\theta(t)|^2, \tag{12-a}$$

$$I_B(t) = 0, \tag{12-b}$$

$$I_C(t) = |\sin\theta(t)|^2. \tag{12-c}$$

As long as the adiabatic condition is well satisfied, the initial excitation in cavity **A** can robustly transfer to **C** without establishing an appreciable intermediate amplitude in cavity **B**.

By contrast, for the initial condition with $|\psi(0)\rangle = |\varphi_C\rangle$, according to Eq. (11), the coefficients for the eigenstates are $a_0(0) = 0$ and $a_\pm(0) = \sqrt{2}/2$, meaning the state function evolves as the in-phase superposition of $|\psi_+(t)\rangle$ and $|\psi_-(t)\rangle$. By defining $\alpha(t) = \int_0^t \varepsilon_+(t')dt'$ and ignoring the dissipation of the system, the transient state function can be written as

$$|\psi(t)\rangle = \sin\theta(t)\cos\alpha(t)|\varphi_A\rangle - i\sin\alpha(t)|\varphi_B\rangle + \cos\theta(t)\cos\alpha(t)|\varphi_C\rangle. \tag{13}$$

Thus, the sound intensities in the cavities with time $t$ are

$$I_A(t) = |\sin\theta(t)|^2|\cos\alpha(t)|^2, \tag{14-a}$$

$$I_B(t) = |\sin\alpha(t)|^2, \tag{14-b}$$

$$I_C(t) = |\cos\theta(t)|^2|\cos\alpha(t)|^2. \tag{14-c}$$

In addition to $\theta(t)$, according to Eq. (14), we know the final sound distributions in the three cavities relate to $\alpha(t)$. Specifically, when $\alpha(t \to +\infty) = (n+1/2)\pi$ with $n$ being the integer, we have $I_B(t \to \infty) = 1$ after the modulation, meaning the sound wave initially excited in **C** finally transfers to **B**, rather than **A**.

For the initial condition with $|\psi(0)\rangle = |\varphi_B\rangle$, with the similar analysis as used above, we find the coefficients for the three eigenstates $a_0(0) = 0$ and $a_\pm(0) = \pm\sqrt{2}/2$, meaning that, in the adiabatic limit, the state function evolves as the out-of-phase superposition of $|\psi_+(t)\rangle$ and $|\psi_-(t)\rangle$. Thus, we get the transient state function

$$|\psi(t)\rangle = -i\sin\theta(t)\sin\alpha(t)|\varphi_A\rangle + \cos\alpha(t)|\varphi_B\rangle - i\cos\theta(t)\sin\alpha(t)|\varphi_C\rangle, \tag{15}$$

and the sound intensities in the cavities with time $t$ is

$$I_A(t) = |\sin\theta(t)|^2|\sin\alpha(t)|^2, \tag{16-a}$$

$$I_B(t) = |\cos\alpha(t)|^2, \tag{16-b}$$

$$I_C(t) = |\cos\theta(t)|^2|\sin\alpha(t)|^2. \tag{16-c}$$

With the same condition $\alpha(t \to +\infty) = (n+1/2)\pi$, apparently, the initially excited sound wave in cavity **B** transfers to **A**.

## Data availability

The main data supporting the findings of this study are available within this letter and its supplementary information. The source data generated in this study have been deposited in Figshare repository https://doi.org/10.6084/m9.figshare.25157849. Source data are provided in this paper. Source data are provided with this paper.

## Code availability

The code used to analyze the data and generate the plots for this paper is available from the corresponding author upon request.

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

## Acknowledgements

We thank Prof. Klaas Bergmann for discussing the specific aspects of STIRAP and for valuable suggestions in the process of finalizing the manuscript. This work was supported by the National Key R&D Program of China (Grants No. 2023YFA1406900 (Z.-G.C.) and 2022YFA1405000 (Y.-Q.L.)), the National Natural Science Foundation of China (Grant Nos. 12304492 (Y.-G.P.), 11690030 (X.-F.Z) and 11690032 (X.-F.Z)), the Natural Science Foundation of Jiangsu Province, Major Project (No. BK20212004 (Y.-Q.L.)), the Innovation Program for Quantum Science and Technology (No. 2021ZD0301500 (P.C.)), the China Postdoctoral Science Foundation (No. 2023M731609 (Z.-X.C.)), and Jiangsu Funding Program for Excellent Postdoctoral Talent (No. 2023ZB473 (Z.-X.C.)).

## Author contributions

Z.-X.C. and Y.-G.P. contributed equally to this work. Z.-X.C. and X.-F.Z. conceived the idea. Z.-X.C., Y.-G.P. and Z.-G.C. developed the theory. Z.-X.C., Y.L. and P.C. performed the experiment. All authors contributed to analyzing the data and writing the manuscript. Y.Q.L. supervised the project.

## Competing interests

The authors declare no competing interests.
