## [Peer Review File · Nature Communications]

Robust temporal adiabatic passage with perfect frequency conversion between detuned acoustic cavitiesReviewer #1 (Remarks to the Author):

In this manuscript, the authors introduce a route toward complete acoustic energy transfer between cavity modes oscillating at dissimilar frequencies and demonstrate the concept experimentally. The basic idea is the introduction of temporal modulation in the electrically-controlled coupling to erase temporal phase mismatching between detuned cavities. This process, termed adiabatic passage, was already known (refs. [11], [12]) but the authors go a step further and perform an experimental demonstration using acoustic resonators. Notably, the agreement between the developed theory and the experiments is quite remarkable. This highlights the validity and robustness of the data interpretation and conclusions as reported in the manuscript.

In my opinion, this is a highly relevant step in the field and, therefore, deserves publication in Nat. Comm., mainly as a consequence of this first (as far as I know) experimental demonstration. The data analysis and interpretation of the results are correct as far as I can see. The methodology is sound, and the Suppl. Material documents include details enough to reproduce the results.

I only have a comment that the authors should address to improve the manuscript's quality and enhance its multidisciplinary aspect, as required in this journal. Probably, the current system fused in the experiments cannot find practical applications, but I understand that the same concept could be extended to nano-electromechanical systems (NEMS) and even optomechanical cavities (though in this last case, the manipulation of the cavities should be done optically). Such systems are quite interesting from a practical perspective, as they can be fabricated in chips and be used in applications in both the classical and quantum realms. So I think that the authors should comment on how to extend their findings to such relevant technological platforms since they can also support multiple mechanical resonances (see some recent works in the field of cavity optomechanics involving different mechanical modes: Nature 537, 80–83 (2016); Phys. Rev. Lett., 127, 073601 (2021); Nature 606, 82–87 (2022)) and it could be interesting to transfer energy between them using this new technique.

Reviewer #2 (Remarks to the Author):

I spent a long time reviewing this manuscript. Unfortunately, it suffers from a severe lack of clarity and coherence in its presentation. In particular, I did not see the significance and impact of this work. The reason is twofold. Firstly, the concept of stimulated-Raman adiabatic passage is not novel, as there have been numerous experiments demonstrating similar effects, albeit on different experimental platforms. Secondly, the key techniques mentioned in this manuscript have already been implemented in previous experiments, including some conducted by the authors' own group. Consequently, I cannot recommend this manuscript for publication in Nature Communications, a renowned and esteemed journal. Below, I provide specific comments primarily regarding the presentation.

(1) The first sentence in the abstract. "In acoustics, efficient directional transfer... requires identical energy levels, otherwise distinct energy levels..." This sentence is quite obscure, even for the colleagues in the field of acoustics. The concepts are mixed and unclear, too. The author should rephrase it using more straightforward language.

(2) The third sentence in the abstract. "...here we introduce temporal modulation (periodically flipped couplings assisted by electrically-controlled coupling assisted by acoustic-elastic-electronic interaction) to..." This sentence is very convoluted and difficult to read.

(3) Figure 1 illustrates the mechanism of nonreciprocal adiabatic passage. It not easy to see the connection between Figs. 1a and 1b. For example, the emulation between the electromagnetic pulses (S_t and P_t) and the dynamic couplings should be elaborated carefully.

(4) Figure captions should be presented more carefully. For example, the curves in Fig. 3c (similarly, Figs. 4c, 4e, 5c, 5e) should be described in details.

Only a few examples are provided above. The manuscript should be revised substantially before submitting somewhere.

Reviewer #3 (Remarks to the Author):

This work investigates frequency conversion and energy transfer between detuned resonator via temporally modulated coupling introduced by electronic circuit. I find it's interesting since they show efficient energy conversion. However, they use their own definition of energy transfer efficiency, so I am not sure if it's really efficient energy transfer. Also, demonstration of circulator and one-way perfect absorbers is shown, but they can be realized by other ways and the demonstrations don't really emphasize their key findings (I think it would be better to focus on physics and physical analogy). I may reconsider after revision.

Here are my comments

- 1.The term "transfer efficiency" generally refers to the ratio of transferred energy to input energy. In this work, it seems to be defined as the energy of one cavity resonator relative to the other at a particular moment in time. Using this definition, it's conceivable to approach 100% energy transfer. Yet, a more comprehensive definition might consider the "input energy" of one cavity (prior to $t=0$) against the aggregate output energy of the alternate resonator at a subsequent time point. Accounting for this lag might yield efficiencies of less than 10%, especially once decay is factored in.
- 2.While the current model focuses on a singular excitation for the cavity resonator followed by modulated coupling, I'm curious about its behavior under a continuous excitation scenario. Can the same efficiencies and principles be maintained?
- 3.The depiction of three resonators in Figure 1 can be a tad misleading. Even though the experimental setup and mathematical equations suggest no direct physical coupling between resonators A and C, the figure should be clearer in representing that any connections between A and C result from their respective connections to B.
- 4.The observed nonreciprocity of three cavity resonators with gaussian-envelop modulation, while interesting, isn't particularly surprising. Such modulated coupling techniques are common for resonator arrays, and their application often leads to the realization of nonreciprocity.
- 5.The manuscript emphasizes the efficient energy transfer between detuned resonators, but many of these phenomena, like beating, can also be seen in standard systems, such as coupled identical resonators. Using these standard approaches, one could also realize circulation and one-way absorption.
- 6.While the demonstrations of circulators and one-way sound absorbers are commendable, such phenomena can be shown through other means. If the paper's core objective revolves around the physics and physical analogies, then maybe there's a need to steer away from applications. For instance, Figure 2a showcases bidirectional electronic couplings. By using a singular electronic coupling, the study could elucidate nonreciprocity, possibly achieving true 100% energy conversion, even with detuned resonators.

List of Major Revisions (Manuscript: NCOMMS-23-29692-T)

We want to thank the reviewers for their valuable comments and suggestions. Their feedback has been instrumental in improving our manuscript. In our resubmitted manuscript, we have made revisions accordingly, and the changes are marked in blue. Below, we summarize the major revisions.

1, We have thoroughly revised the whole manuscript with the help from Dr. Klaas Bergmann. In particular, the abstract and the conclusion are carefully rephrased to deliver the main contributions of our work. We also modified the title to emphasize that this work aims for *detuned acoustic cavities*.

2, We have simplified Fig. 1 to present the idea of our temporal quasi-phase matching and the generalized STIRAP technology for sound waves. In addition, Figs. 2-5 are also revised, and the related captions are thoroughly modified for clarity.

3, We added a clear definition of the wave transfer efficiency $|S_{BA}(t)|^2$ in Eq. (2). New experimental results are added in supplementary Section II to decrease the cavity damping. New simulation results are added in the supplementary Section IV for discussing the transferred energy after the mode evolution.

4, We clarified the novelty of the non-reciprocal function of our proposal by comparing it with existing solutions for airborne sound, which is strengthened in the main text.

5, We also discussed extending our work to other realms, such as nano-electromechanical and optomechanical systems. This is given in the conclusion part.

Point-to-point response to Reviewers' comments

Reviewer #1 (Remarks to the Author)

In this manuscript, the authors introduce a route toward complete acoustic energy transfer between cavity modes oscillating at dissimilar frequencies and demonstrate the concept experimentally. The basic idea is the introduction of temporal modulation in the electrically-controlled coupling to erase temporal phase mismatching between detuned cavities. This process, termed adiabatic passage, was already known (refs. [11], [12]), but the authors go a step further and perform an experimental demonstration using acoustic resonators. Notably, the agreement between the developed theory and the experiments is quite remarkable. This highlights the validity and robustness of the data interpretation and conclusions as reported in the manuscript.

In my opinion, this is a highly relevant step in the field and, therefore, deserves publication in Nat. Comm., mainly as a consequence of this first (as far as I know) experimental demonstration. The data analysis and interpretation of the results are correct as far as I can see. The methodology is sound, and the Suppl. Material documents include details enough to reproduce the results.

Response: We thank the reviewer for these very positive comments, which are fully in line with our intention when preparing the manuscript.

I only have a comment that the authors should address to improve the manuscript's quality and enhance its multidisciplinary aspect, as required in this journal. Probably, the current system used in the experiments cannot find practical applications, but I understand that the same concept could be extended to nano-electromechanical systems (NEMS) and even optomechanical cavities (though in this last case, the manipulation of the cavities should be done optically). Such systems are quite interesting from a practical perspective, as they can be fabricated in chips and be used in applications in both the classical and quantum realms. So I think that the authors should comment on how to extend their findings to such relevant technological platforms since they can also support multiple mechanical resonances (see some recent works in the field of cavity optomechanics involving different mechanical modes: Nature 537, 80–83 (2016); Phys. Rev. Lett., 127, 073601 (2021); Nature 606, 82–87 (2022)) and it could be interesting to transfer energy between them using this new technique.

Response: We thank the reviewer for these stimulating and interesting suggestions. Our developed temporal quasi-phase matching (TQPM) method and generalized STIRAP technology constitute a universal and versatile platform for transient sound wave steering and can enrich the toolbox of nonreciprocal sound devices. We agree with the reviewer that these achievements can also shed lights on the state manipulation in nano-electromechanical systems (NEMS) and even optomechanical systems.

Since this manuscript mainly focuses on establishing the STIRAP concept in aeroacoustics, in the discussion part of the resubmission, we added the following words to enhance the multidisciplinary aspect:

In addition, the developed TQPM theory and the generalized STIRAP methodology can shed lights on the state manipulation in nano-electromechanical as well as optomechanical systems, which can support multiple mechanical resonances and find more applications in both the classical and quantum realms^{30,45,46}.

Reviewer #2 (Remarks to the Author)

I spent a long time reviewing this manuscript. Unfortunately, it suffers from a severe lack of clarity and coherence in its presentation. In particular, I did not see the significance and impact of this work. The reason is twofold. Firstly, the concept of stimulated-Raman adiabatic passage is not novel, as there have been numerous experiments demonstrating similar effects, albeit on different experimental platforms. Secondly, the key techniques mentioned in this manuscript have already been implemented in previous experiments, including some conducted by the authors' own group. Consequently, I cannot recommend this manuscript for publication in Nature Communications, a renowned and esteemed journal. Below, I provide specific comments primarily regarding the presentation.

Response: We thank the reviewer for these suggestive comments, even though they are critical. Regarding the presentations, we appreciate the reviewer's relevant suggestions, and we have made efforts to thoroughly revise the manuscript. The following responses are about detailed revisions.

In terms of the significance of this work, we gave the following clarification, hoping to overcome the reviewer's concern. Firstly, it is correct that the concept of transfer between three entities based on "counter-intuitively" delayed interactions is not new. However, we need to point out that the original concept, published in 1990, was taken up by many groups worldwide, and was applied in many different contexts and fields with dozens of publications, *e.g.*, in *Science*, *Nature*, and *Phys. Rev. Lett.* In particular, in light of the numerous advantages, including the robustness and non-reciprocity, the methodology of STIRAP was generally applied in *optical waveguide* systems to realize robust and asymmetric (without breaking the reciprocity) wave coupling. For acoustics, as far as we know, it should be the first time for us to bring the STIRAP into acoustic [Phys. Rev. Lett., **122**, 094501 (2019)] in a static and linear platform without breaking the reciprocity.

In atom physics, photon energies are adjusted to guarantee energy conservation in the transfer of population between quantum states in distinct energy levels. However, such a mechanism is absent in acoustics. So far, the STIRAP implementations in both optics and acoustics are restricted to bulky waveguide systems with identical entities, where the frequency conversion is absent. In addition, the waveguide systems are static in nature and do not break the time-reversal symmetry, rendering them essentially the reciprocal functional devices.

With such a background, in this work, we develop effective strategies for realizing time-modulated couplings and gain-enhanced acoustic cavities, with which we have generalized the STIRAP to the time domain with arbitrarily detuned cavities. **As Referee #1 judged, this is the first experimental demonstration and is a highly relevant step in this field.** In addition to the robust forward wave energy relocation, we furtherly propose several non-reciprocal devices, such as the circulator and one-way absorber, by utilizing different adiabatic passages. Thus, the newly reported acoustic adiabatic evolution in this work not only enriches the transient steering for sound, but also provides new insights into elastic and nano-electromechanical systems.

(1) The first sentence in the abstract. "In acoustics, efficient directional transfer... requires identical energy levels, otherwise distinct energy levels..." This sentence is quite obscure, even for the colleagues in the field of acoustics. The concepts are mixed and unclear, too. The author should rephrase it using more straightforward language.

(2) The third sentence in the abstract. "...here we introduce temporal modulation (periodically flipped couplings assisted by electrically-controlled coupling assisted by acoustic-elastic-electronic interaction) to..." This sentence is very convoluted and difficult to read.

Response: We agree with the reviewer on these two points that parts of the abstract need improvement. Following is our rephrased abstract by using more straightforward language to describe the contributions of our work:

*Abstract: Many phase-sensitive processes, including frequency conversion in nonlinear optics, require phase matching to enable efficient energy transfer. Phase matching can be achieved by various methods, such as using birefringent crystals and quasi-phase matching in periodic structures. In this work, we show how phase matching can be leveraged for transient acoustic wave frequency conversion in a linear system. We propose a temporal quasi-phase matching method and realize robust and complete acoustical energy transfer between cavity modes with different resonant frequencies. In a set of three detuned cavities, **A**, **B**, and **C**, the time-varying coupling is established between **A** and **B** as well as **B** and **C**. Analogy to the concept of stimulated Raman adiabatic passage in atomic physics, amplitudes of the two electrically controlled couplings are modulated as time-delayed Gaussian functions with **B-C** coupling preceding **A-B** coupling. Furthermore, the couplings' signs are periodically flipped to eliminate temporal phase mismatching between waves in the detuned cavities. As a result, acoustic energy, initially deposited in cavity **A**, is transferred to cavity **C** without appreciable excitation of the intermediate cavity **B**. The robustness against variations of the coupling parameters is demonstrated. We further demonstrate this design's non-reciprocal frequency conversion properties of our design. Our research takes a pivotal step towards expanding wave steering through time-dependent modulations and is promising to extend the frequency conversion based on state evolution in various linear Hermitian systems to nonlinear and non-Hermitian regimes.*

(3) Figure 1 illustrates the mechanism of nonreciprocal adiabatic passage. It not easy to see the connection between Figs. 1a and 1b. For example, the emulation between the electromagnetic pulses (S_t and P_t) and the dynamic couplings should be elaborated carefully.

Response: We appreciate the reviewer for this valuable comment. To present the emulation between the electromagnetic pulses and the dynamic couplings clearer, in the revised manuscript, we have simplified Fig. 1 (shown here as Fig. R1) and rephrased the caption as follows.

Fig. R1 | Robust and nonreciprocal adiabatic passage. *a*, Stimulated Raman adiabatic passage in a three-level system (left panel). With a counterintuitive electromagnetic pulse sequence that $S(t)$ precedes $P(t)$ with Δt (shown in the right panel with the vertical coordinate for electric field E), the population in $|1\rangle$ can robustly transfer to $|3\rangle$ without transient population in $|2\rangle$. *b*, A schematic for the transient acoustic adiabatic passages with three detuned cavities resonant at f_A , f_B and f_C , respectively (left panel). In addition to the Gaussian-shaped envelopes, the signs of the couplings $\kappa_{AB}(t)$ and $\kappa_{BC}(t)$ are periodically flipped to realize temporal quasi-phase mismatching between the waves in adjacent cavities (right panel). The sequential dynamic couplings bring a robust transfer from A to C and can enable our system to be a circulator or unidirectional absorber for transient waves, with the detailed results presented in Figs. 3-5.

(4) Figure captions should be presented more carefully. For example, the curves in Fig. 3c (similarly, Figs. 4c, 4e, 5c, 5e) should be described in details.

Response: We thank the reviewer for the careful reading and valuable suggestion. All the figure captions and related main text have been polished, as marked in blue in the resubmission.

Only a few examples are provided above. The manuscript should be revised substantially before submitting somewhere.

Response: In addition to these examples mentioned above, we have further carefully and thoroughly revised the whole manuscript and the supplementary materials.

Reviewer #3 (Remarks to the Author)

This work investigates frequency conversion and energy transfer between detuned resonator via temporally modulated coupling introduced by electronic circuit. I find it's interesting since they show efficient energy conversion. However, they use their own definition of energy transfer efficiency, so I am not sure if it's really efficient energy transfer. Also, demonstration of circulator and one-way perfect absorbers is shown, but they can be realized by other ways and the demonstrations don't really emphasize their key findings (I think it would be better to focus on physics and physical analogy). I may reconsider after revision.

Response: We appreciate the reviewer for the favorable judgement that "*I find it's interesting since they show efficient energy conversion*" and for these comments to improve our manuscript. The reviewer is mainly concerned about the definition of "energy transfer efficiency" and the nonreciprocal functionalities, which are thoroughly clarified in the following responses to the questions 1 and 6, respectively. We hope our revisions can solve these concerns.

Here are my comments

1. The term "transfer efficiency" generally refers to the ratio of transferred energy to input energy. In this work, it seems to be defined as the energy of one cavity resonator relative to the other at a particular moment in time. Using this definition, it's conceivable to approach 100% energy transfer. Yet, a more comprehensive definition might consider the "input energy" of one cavity (prior to $t=0$) against the aggregate output energy of the alternate resonator at a subsequent time point. Accounting for this lag might yield efficiencies of less than 10%, especially once decay is factored in.

Response: The reviewer raised an important question about the definition of transfer efficiency. Considering the uniform loss of the cavities and the Hermiticity of the mutual coupling, it is apparent that the sound energy of the system damps with time and, as the reviewer said, the energy transfer efficiency cannot be unity if compared with the input energy at $t = 0$ rather than that after evolution. Damping might be a universal problem for all passive mechanical systems (e.g., see *Nature* **537**, 80-83 (2016) and *Nature* **606**, 82-87 (2022) recommended by Reviewer #1). Thus, to focus on the wave transfer enabled with mode evolution, we note that it is practical to ignore the system's damping

and define the transfer efficiency as energy proportion after the evolution. For example, in *Nature* **537**, 80-83 (2016), which is about the topological transient energy transfer in the optomechanical system, the transfer efficiency is defined as $E = |c_b(\tau)|^2 / [|c_a(\tau)|^2 + |c_b(\tau)|^2]$ (see the third page of the paper for details).

To avoid misunderstanding, in the resubmission, we give a clear definition for wave transfer efficiency in Eq. (2) (on page 6):

We ignore the damping of the cavities and define the energy transfer efficiency from cavity A to B as

$$|S_{BA}(t)|^2 = |p_B(t)|^2 / \sum_j |p_j(t)|^2, \quad (2)$$

which is practically adopted for investigating state evolution in mechanical systems³⁰. By analyzing the temporally recorded sound waves in Fig. 2b, the maximum of $|S_{BA}(t)|^2$ is only 0.11, consistent with the prediction of $4\kappa_0^2 / (\Omega_{AB}^2 + 4\kappa_0^2)$. In addition, the transfer efficiency varies with a period of $T_0 = 18.6$ ms, which is determined by the Rabi frequency⁵, namely $T_0 = 0.5 / \sqrt{\Omega_{AB}^2 + 4\kappa_0^2}$.

On page 9, the definition of $|S_{CA}|^2$ for STIRAP is clearly given:

Similar to the definition in Eq. (2), we get the forward wave transfer efficiency as $|S_{CA}|^2 = |p_C(t_{\text{end}})|^2 / \sum_j |p_j(t_{\text{end}})|^2$ at $t_{\text{end}} = t_0 + 2\sigma + \Delta t$. Obviously, $|S_{CA}|^2$ denotes the fidelity of the zero-energy mode after the evolution.

On the other hand, starting with the input energy at $t = 0$, we define the total energy transfer efficiency as $|\widetilde{S}_{CA}|^2 = |p_C(t_{\text{end}})|^2 / \sum_j |p_j(t = 0)|^2 = |S_{CA}|^2 e^{-2\Gamma t_{\text{end}}}$, where Γ is the damping rate of the cavities. $|\widetilde{S}_{CA}|^2$ can be effectively improved twofold, *i.e.*, decrease the cavity damping Γ by introducing more gain and shorten the modulation time t_{end} by enlarging the coupling amplitudes.

To prove this, we simulate $e^{-2\Gamma t_{\text{end}}}$ as a function of Γ and $\tilde{\kappa}_0$, which is the maximum of the effective coupling (see Fig. S7 in supplementary Section IV for other parameters). Consistent with our prediction, Fig. R2a shows that $e^{-2\Gamma t_{\text{end}}}$ can be effectively enlarged by decreasing the damping rate Γ and increasing $\tilde{\kappa}_0$. Figures R2b-d show the simulated wave transfer with $\tilde{\kappa}_0 = 6.1$ Hz and $\Gamma = 0.11$ Hz (the red star in Fig. R2a), with which the total energy transfer efficiency can be up to 0.5. **These simulation results were added in the Supplementary section IV (marked in blue).**

Fig. R2 **a**, Calculated values of $e^{-2\Gamma t_{end}}$ as a function of $\tilde{\kappa}_0$ and Γ . The white line denotes the contour of 0.5, and the red star indicates the parameters of $\tilde{\kappa}_0 = 6.1$ Hz and $\Gamma = 0.11$ Hz used in **b-d**. **b-d**, Simulated sound wave transfer from cavities A to C (**b**), C to B (**c**), and B to A (**d**). The Gaussian-shaped couplings are omitted here.

We stress that the cavity damping Γ can be further reduced to such a low level by updating the gain circuit with a voltage-controlled amplifier (VCA), which is also used in the coupling circuit. This allows precise digital control the amplitude of the gain. In addition, as shown in Fig. R3a, a phase shifter is used to ensure exactly in-phase gain. By digitally controlling the circuit, the measured (circles) and fitted (curves) exciting spectra in Fig. R3b show that the cavity's damping rate can decrease from the original $\Gamma = 10$ Hz (black) to 0.8 Hz (blue) or even 0.1 Hz (red). **These results were updated in the Supplementary section II (marked in blue).**

Fig. R3 **a**, A schematic view of the experimental setup that introduces exact pure gain by adopting the phase shifter and the VCA into the feedback circuit. **b**, The measured (circles) and fitted (curves) excitation spectra of a cavity with different gain strengths. The damping rate decreases from the original $\Gamma = 10$ Hz (black) to 0.8 Hz (blue) or even 0.1 Hz (red).

In the resubmission, we added the following discussion on page 10 of the main text: *In addition, though the parameters realized here are suitable for the demonstration purposes, the damping of the cavities can be further precisely reduced to increase the transferred sound energy (see Supplementary sections II and IV for the experiment and simulation results).*

2. While the current model focuses on a singular excitation for the cavity resonator followed by modulated coupling, I'm curious about its behavior under a continuous excitation scenario. Can the same efficiencies and principles be maintained?

Response: This is a very good question that also puzzled us in the early stage of the work, but the answer is NO. The energy transfer studied in this work is related to the state evolution, and the governing equation is $i \frac{d}{dt} |\psi(t)\rangle = \mathbf{H} |\psi(t)\rangle$, namely the source is turned down during the modulation. Comparatively, if the sound source is kept on, the governing equation becomes $i \frac{d}{dt} |\psi(t)\rangle = \mathbf{H} |\psi(t)\rangle + |s(t)\rangle$.

Fig. R4 **a**, Complete and robust sound wave transfer from cavity **A** to **C**. The dynamic couplings (not shown here) take place after $t = 0$, and the sound source (represented as the white arrow) only works in the cavity **A** for $t < 0$. **b**, The same as **(a)** but with the sound source being kept on during the whole process.

As a demonstration, we simulate the wave dynamics in the three-cavity system with the couplings being modulated after $t = 0$. A continuous source at cavity **A** can

be represented as $|s(t)\rangle = \begin{bmatrix} 1 \\ 0 \\ 0 \end{bmatrix} e^{-i\omega t}$ with $\begin{bmatrix} 1 \\ 0 \\ 0 \end{bmatrix}$ being the initial zero-energy

eigenstate. When the dynamic couplings take place for $t > 0$, the zero-energy eigenstate varies accordingly, and the continuous source $|s(t)\rangle$ excites the combination of all the eigenstates, making the wave dynamics complicated. Figure R4a shows the simulated sound waves in the system with the sound source (denoted with the white arrow) being switched off after $t = 0$. The sound energy transfer from cavity **A** to **C** clearly shows the evolution of the zero-energy eigenstate. However, when the

sound source is kept on all the time, as shown in Fig. R4b, the wave behaviors in the system become complicated since all three eigenmodes of the system are excited during the modulation. **Thus, the adiabatic passages only work for initial excitations, and the damping of the cavities inevitably dissipates the wave energy during the state evolution.**

To avoid misunderstanding, we added the following discussions on page 10 of the resubmission:

Notably, the robust sound energy transfer results from the adiabatic evolution of the initially prepared eigenstate $|\psi_0\rangle$, thus the sound source must be turned off before dynamic couplings take place.

3.The depiction of three resonators in Figure 1 can be a tad misleading. Even though the experimental setup and mathematical equations suggest no direct physical coupling between resonators A and C, the figure should be clearer in representing that any connections between A and C result from their respective connections to B.

Response: We thank the reviewer for this valuable comment. We have carefully revised Fig. 1 and the caption to make it clearer (please see our response to the third comment of Reviewer #2).

4.The observed nonreciprocity of three cavity resonators with gaussian-envelop modulation, while interesting, isn't particularly surprising. Such modulated coupling techniques are common for resonator arrays, and their application often leads to the realization of nonreciprocity.

Response: Non-reciprocal wave steering is a long-lasting and challenging topic for the acoustic community. To our knowledge, so far, there are mainly two strategies proposed to realize sound circulators (see Refs. 34-35 of the main text). As summarized in Table R1, one strategy is to introduce the biased flow, which is quite hard to control; another recipe is to synthesize the angular momentum, *i.e.*, design spatial-temporal modulation. **Most importantly, these two designs rely on cavity resonance or Floquet harmonic wave generations and are both invalid for pulsed sound waves.** Figure R5 gives the related experimental design.

Table R1 Comparisons of the sound circulators

Strategy for sound circulators	Biased flow	Synthetic angular momentum	Adiabatic passage (our work)
Floquet time Modulation	×	√	×
Adiabatic time evolution	×	×	√
Applicable for transient sound waves	×	×	√

Applicable for continuous excitation	√	√	×
---	---	---

Comparatively, our work with the adiabatic evolution provides a distinct mechanism to effectively realize a circulator for transient sound waves. We point out that although the STIRAP methodology has been utilized in optical and acoustic waveguide platforms, these implementations are static and are reciprocal in nature (see Section A3 of our review paper *J. Phys. B* **52**, 202001, (2019) for details).

Fig. R5 **a**, Sound circulator realized with biased flow (adapted from Ref. 34 of the main text, *Science* **343**, 516, (2014)). **b**, Sound circulator with synthetic angular momentum to break the non-reciprocity (adapted from Ref. 35 of the main text, *Nat. Commun.* **7**, 11744, (2016)).

In the resubmission, we clarified the novelty of our circulator on page 12: *Comparatively, without any biased fluid flow or synthetic angular momentum, the adiabatic acoustic passages with dynamic couplings provide a distinct mechanism to realize circulators, particularly for transient sound waves*³⁴⁻³⁸.

5. The manuscript emphasizes the efficient energy transfer between detuned resonators, but many of these phenomena, like beating, can also be seen in standard systems, such as coupled identical resonators. Using these standard approaches, one could also realize circulation and one-way absorption.

Response: It is well-known that the coupling assists unitary energy transfer between identical resonators (or waveguides), which share identical resonant frequencies (or propagation constants). **We experimentally realized the beating and adiabatic passages for identical acoustic cavities, which were presented in Supplementary Section VI.**

However, in this paper, we aim for the **detuned resonators**. Due to the phase-mismatching between the detuned resonators, beating with unitary energy transfer becomes unachievable. To solve this problem and realize robust unitary energy transfer between arbitrarily detuned cavities, here we came up with the TQPM strategy, which is implemented in acoustic resonators by periodically switching the coupling phase between in-phase and out-of-phase. On this basis, we generalized the concept of

STIRAP and realized robust energy transfer and frequency conversion between the detuned cavities. In addition, by optimizing the time lag between the dynamic couplings, we proposed a novel and effective mechanism to realize a nonreciprocal circulator for pulsed sound energy. Notably, the TQPM has general consequences for designing adiabatic passages between arbitrarily detuned cavities, which is extensively discussed in Supplementary Section VII with simulations.

Thus, we hope the reviewer will see the advancement and generality of our work.

6. While the demonstrations of circulators and one-way sound absorbers are commendable, such phenomena can be shown through other means. If the paper's core objective revolves around the physics and physical analogies, then maybe there's a need to steer away from applications. For instance, Figure 2a showcases bidirectional electronic couplings. By using a singular electronic coupling, the study could elucidate nonreciprocity, possibly achieving true 100% energy conversion, even with detuned resonators.

Response: We appreciate the reviewer's positive comment that "*the demonstrations of circulators and one-way sound absorbers are commendable*". As summarized in Table R1, our work develops a distinct mechanism to realize sound circulator, particularly for transient sound waves.

In addition, we agree with the reviewer that nonreciprocity can be easily realized in our platform by designing unidirectional electronic coupling. However, we cannot agree that "*possibly achieving true 100% energy conversion, even with detuned resonators*". In fact, by simply introducing unidirectional coupling, we cannot achieve effective energy conversion between detuned resonators. Without loss of generality, we set the detuning of cavities **A** and **B** as $\Omega_{AB} = |f_A - f_B| = 70$ Hz and simulate the wave dynamics in the two-cavity system with cavity **A** being initially prepared.

Fig. R6 Simulated sound waves in a two-cavity system with detuning to be $\Omega_{AB} = |f_A - f_B| = 70$ Hz. Cavity **A** is with the initial condition of $p_A(t=0) = 1$ **a**, The case with unidirectional and static coupling $\kappa_0 = 9$ Hz from cavity **A** to **B**. **b**, The unidirectional coupling is periodically flipped as $\kappa_{AB}(t) = \kappa_0 \text{sign}[\cos(2\pi\Omega_{AB}t)]$. **c**, The case with mutual and time-varying coupling $\kappa_{AB}(t) = \kappa_{BA}(t)$.

First, when there is only static and unidirectional coupling $\kappa_0 = 9$ Hz from cavity **A** to **B**, the Hamiltonian of the system should be $H = 2\pi \begin{bmatrix} f_A - i\Gamma & 0 \\ \kappa_0 & f_B - i\Gamma \end{bmatrix}$. Apparently, this Hamiltonian is non-Hermitian. As shown in Fig. R6a, the simulated results show that the initially prepared wave energy in cavity **A** cannot be transferred to **B** at all, and the “*true 100% energy conversion*” is impossible. In addition, due to the phase mismatching, the newly generated sound energy in cavity **B** is limited, fluctuating with time. When the unidirectional coupling is time-modulated according to our TQPM theory, namely $\kappa_{AB}(t) = \kappa_0 \text{sign}[\cos(2\pi\Omega_{AB}t)]$, Fig. R6b shows that the sound energy in cavity **B** builds up first and then decrease together with the energy in **A**. Finally, we introduce mutual and dynamic couplings to the system, and the Hamiltonian of the system should be $H = 2\pi \begin{bmatrix} f_A - i\Gamma & \kappa_{BA}(t) \\ \kappa_{AB}(t) & f_B - i\Gamma \end{bmatrix}$ with $\kappa_{AB}(t) = \kappa_{BA}(t)$. Now, the couplings are Hermitian and do not introduce extra energy to the system. Figure R6c shows that the sound energy in the system oscillates between the two cavities with the wave transfer efficiency $|S_{BA}|^2$ be unity at some discrete time points.

In summary, with only a simple model, we can conclude unidirectional coupling implies non-reciprocity but cannot realize complete energy transfer. The conclusion holds for the three-cavity systems. **We hope our arguments will further show the importance of temporal phase matching. Our work provides a general recipe to realize the non-reciprocal sound energy transfer between the detuned cavities by combining our TQPM theory with STIRAP.**

Reviewer #2 (Remarks to the Author):

Substantial revisions are made in the resubmitted manuscript, most of them are satisfied. However, I am afraid I can't agree with the author's conclusion in the response letter "So far, the STIRAP implementations in both optics and acoustics are restricted to bulky waveguide systems with identical entities, where the frequency conversion is absent". In the manuscript (page 7, Line 8), the authors mentioned temporary quasi-phase matching (TQPM) in nonlinear optics. In fact, STIRAP with frequency conversion has also been theoretically proposed and experimentally observed in previous nonlinear optics studies (as shown in the following). I am not sure if the authors were inspired by these works (or were not aware of them), but the principles of these studies are closely related to this article. The authors should give a sufficient discussion in the article and point out the differences and advantages of their work compared with these works. Some references: 1. Adiabatic processes in frequency conversion; 2. Efficient Three-Process Frequency Conversion Based on Straddling Stimulated Raman Adiabatic Passage; 3. Cascaded frequency conversion under nonlinear stimulated Raman adiabatic passage (2021); 4. Synthesis of white laser source based on nonlinear frequency conversion with stimulated Raman adiabatic passage.

Reviewer #3 (Remarks to the Author):

The authors have addressed all of my comments, particularly clarifying the energy conversion efficiency. Additionally, the revised manuscript now reveals that continuous excitation is not favorable for adiabatic energy transfer. Since the responses are thorough, I have no further comments. Therefore, I would recommend the publication of the revised manuscript in Nature Communications.

Point-to-point response to Reviewers' comments

Reviewer #2 (Remarks to the Author):

Substantial revisions are made in the resubmitted manuscript, most of them are satisfied. However, I am afraid I can't agree with the author's conclusion in the response letter "So far, the STIRAP implementations in both optics and acoustics are restricted to bulky waveguide systems with identical entities, where the frequency conversion is absent". In the manuscript (page 7, Line 8), the authors mentioned temporary quasi-phase matching (TQPM) in nonlinear optics. In fact, STIRAP with frequency conversion has also been theoretically proposed and experimentally observed in previous nonlinear optics studies (as shown in the following). I am not sure if the authors were inspired by these works (or were not aware of them), but the principles of these studies are closely related to this article. The authors should give a sufficient discussion in the article and point out the differences and advantages of their work compared with these works. Some references: 1. Adiabatic processes in frequency conversion; 2. Efficient Three-Process Frequency Conversion Based on Straddling Stimulated Raman Adiabatic Passage; 3. Cascaded frequency conversion under nonlinear stimulated Raman adiabatic passage (2021); 4. Synthesis of white laser source based on nonlinear frequency conversion with stimulated Raman adiabatic passage.

Response: We appreciate the reviewer's positive comments that "*Substantial revisions are made in the resubmitted manuscript, most of them are satisfied.*" We express our gratitude to the referee for pointing out the integration of adiabatic passage into nonlinear optics as a means of achieving efficient, scalable broadband frequency conversion. It is important to acknowledge that much of this is still in the theoretical stage. Realizing a nonlinear response in low-amplitude air acoustics poses a significant challenge, and efficient acoustic frequency conversion using spatial (quasi-) phase matching in a nonlinear medium remains unachieved. Consequently, the proposed strategies in acoustics are still a topic of ongoing exploration. However, we concur that this represents a promising and stimulating avenue worthy of attention. We have included additional discussions on this subject as outlined below (marked in blue on page 5):

Significantly, this approach may be considered as the temporal analog to the spatial quasi-phase matching strategy extensively employed in nonlinear optics. This strategy involves meticulously poling nonlinear crystals to counteract phase mismatches among various harmonics³¹⁻³³. By leveraging this technique, one can achieve a broad-spectrum effect, further enhanced through the application of adiabatic passage³⁴⁻³⁷.

Reviewer #3 (Remarks to the Author):

The authors have addressed all of my comments, particularly clarifying the energy conversion efficiency. Additionally, the revised manuscript now reveals that continuous excitation is not favorable for adiabatic energy transfer. Since the responses

are thorough, I have no further comments. Therefore, I would recommend the publication of the revised manuscript in Nature Communications.

Response: We appreciate the reviewer for recommending the publication of our work.